# Self-supervised Adversarial Robustness for the Low-label, High-data Regime

**Sven Gowal\*, Po-Sen Huang\*, Aaron van den Oord, Timothy Mann & Pushmeet Kohli**
DeepMind
London, United Kingdom
{sgowal,posenhuang}@google.com

## Abstract

Recent work discovered that training models to be invariant to adversarial perturbations requires substantially larger datasets than those required for standard classification. Perhaps more surprisingly, these larger datasets can be "mostly" unlabeled. *Pseudo-labeling*, a technique simultaneously pioneered by four separate and simultaneous works in 2019, has been proposed as a competitive alternative to labeled data for training adversarially robust models. However, when the amount of labeled data decreases, the performance of *pseudo-labeling* catastrophically drops, thus questioning the theoretical insights put forward by Uesato et al. (2019), which suggest that the sample complexity for learning an adversarially robust model from unlabeled data should match the fully supervised case. We introduce Bootstrap Your Own Robust Latents (BYORL), a self-supervised learning technique based on BYOL for training adversarially robust models. Our method enables us to train robust representations without any labels (reconciling practice with theory). Most notably, this robust representation can be leveraged by a linear classifier to train adversarially robust models, even when the linear classifier is not trained adversarially. We evaluate BYORL and *pseudo-labeling* on CIFAR-10 and IMAGENET and demonstrate that BYORL achieves significantly higher robustness in the low-label regime (i.e., models resulting from BYORL are up to two times more accurate). Experiments on CIFAR-10 against $\ell_2$ and $\ell_\infty$ norm-bounded perturbations demonstrate that BYORL achieves near state-of-the-art robustness with as little as 500 labeled examples. We also note that against $\ell_2$ norm-bounded perturbations of size $\epsilon = 128/255$, BYORL surpasses the known state-of-the-art with an accuracy under attack of 77.61% (against 72.91% for the prior art).

## 1 Introduction

As neural networks tackle challenges ranging from ranking content on the web (Covington et al., 2016) to autonomous driving (Bojarski et al., 2016) via medical diagnostics (De Fauw et al., 2018), it has becomes increasingly important to ensure that deployed models are robust and generalize to various input perturbations. Unfortunately, despite their success, neural networks are not intrinsically robust. In particular, the addition of small but carefully chosen deviations to the input, called adversarial perturbations, can cause the neural network to make incorrect predictions with high confidence (Carlini & Wagner, 2017a; Goodfellow et al., 2014; Kurakin et al., 2016; Szegedy et al., 2013). Starting with Szegedy et al. (2013), there has been a lot of work on understanding and generating adversarial perturbations (Carlini & Wagner, 2017b; Athalye & Sutskever, 2017), and on building models that are robust to such perturbations (Papernot et al., 2015; Madry et al., 2017; Kannan et al., 2018). Robust optimization techniques, like the one developed by Madry et al. (2017), learn robust models by trying to find the worst-case adversarial examples (by using gradient ascent on the training loss) at each training step and adding them to the training data.

Since Madry et al. (2017), various modifications to their original implementation have been proposed (Zhang et al., 2019; Pang et al., 2020; Huang et al., 2020; Qin et al., 2019). We highlight the simultaneous work from Carmon et al. (2019); Uesato et al. (2019); Zhai et al. (2019a); Najafi et al. (2019) that pioneered the use of additional unlabeled data using *pseudo-labeling*. While, theoretically,

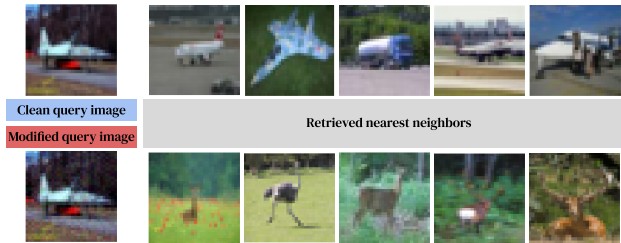

Figure 1: Dangers of using non-robust representation learning. We use a non-robust self-supervised learning technique to learn image representations (i.e., BYOL; Grill et al., 2020). The right-hand side shows CIFAR-10 images closest (in representation space using cosine similarity) to the query image on the left. The top row demonstrates that, when given an unmodified image of an airplane, the nearest matches resemble that query image either visually or semantically. The bottom row demonstrates that a seemingly identical image can be used to retrieve images of animals which are both visually and semantically far from the query image.

robustness can be achieved with only limited amount of labeled data, in practice, it remains difficult to train models that are both robust and accurate in the low-label regime. [1]

Finally, we note that there has been little work towards learning adversarially robust representations that allow for efficient training on multiple downstream tasks (with the exception of Cemgil et al., 2019; Kim et al., 2020). Learning good image representations is a key challenge in computer vision (Wiskott & Sejnowski, 2002; Hinton et al., 2006), and many different approaches have been proposed. Among them state-of-the-art methods include contrastive methods (Chen et al., 2020b; Oord et al., 2018; He et al., 2020) and latent bootstrapping (Grill et al., 2020). However, none of these recent works consider the impact of adversarial manipulations, which can render the widespread use of general representations difficult. As an example, Fig. 1 demonstrates the effect that a non-robust representation has on a content retrieval task, where two seemingly identical query images are matched to widely different images (i.e., their nearest neighbors in representation space).

In this paper, we tackle the issue of learning robust representations that are adversarially robust on multiple downstream tasks in the low-label regime. Our contributions are as follows:

- We formulate Bootstrap Your Own Robust Latents (BYORL), a modification of Bootstrap Your Own Latents (BYOL) (Grill et al., 2020) that enables the training of robust representations without the need for any label information. These representations allow for efficient training on multiple downstream tasks with a fraction of the original labels.
- Most notably, even with only 1% of the labels, BYORL comes close to or even exceeds previous state-of-the-art which uses all labels. For example, for $\ell_2$ norm-bounded perturbations of size $\epsilon = 128/255$ on CIFAR-10, BYORL achieves 75.50% robust accuracy compared to 72.91% for the previous state-of-the-art using all labels. BYORL reaches 77.61% robust accuracy when using all available labels (and additional unlabeled data extracted from 80M-TINYIMAGES; Torralba et al., 2008).
- Finally, we show that the representations learned through BYORL transfer much better to downstream tasks (i.e., downscaled STL-10 (Coates et al., 2011) and CIFAR-100 (Krizhevsky et al., 2014)) than those obtained through *pseudo-labeling* and standard adversarial training. Importantly, we also highlight that classifiers trained on top of these robust representations do not need to be trained adversarially to be robust.

## 2 RELATED WORK

**Adversarial robustness.** Biggio et al. (2013) and Szegedy et al. (2013) observed that neural networks, while they achieve high accuracy on test data, are vulnerable to carefully crafted inputs perturbations, called adversarial examples. Since then, there has been several work on building stronger adversarial examples as well as defense against such adversarial examples (Carlini & Wagner, 2017b; Athalye & Sutskever, 2017; Goodfellow et al., 2014; Papernot et al., 2015; Madry

---

[1]In Uesato et al. (2018) and Carmon et al. (2019), robust accuracy drops by 10% when limiting the number of labels to about 10%.

et al., 2017; Kannan et al., 2018). Arguably, the most successful approach for learning adversarially robust models is adversarial training as proposed by Madry et al. (Athalye et al., 2018; Uesato et al., 2018). This classic version of adversarial training has been augmented in different ways – with changes in the attack procedure (e.g., by incorporating momentum; Dong et al., 2017), loss function (e.g., logit pairing; Mosbach et al., 2018) or model architecture (e.g., using attention; Zoran et al., 2020). We also highlight Zhang et al. (2019), who proposed TRADES which balances the trade-off between standard and robust accuracy. By construction, to the contrary of our proposed method, all aforementioned approaches use label information and are not capable of learning generic representations that might be useful to multiple downstream tasks.

**Semi- and self-supervised learning.** Since human annotations can be expensive, semi- and self-supervised learning approaches that leverage both labeled and unlabeled data have been proposed to improve model performance (Chapelle et al., 2009; Bachman et al., 2014; Berthelot et al., 2019; Laine & Aila, 2017; Miyato et al., 2018; Sajjadi et al., 2016; Xie et al., 2019). A common approach is to train networks to solve a manually-predefined pretext task (e.g., predicting the relative location of image patches) for representation learning, and later use the learned representation for a specific supervised learning task (Dosovitskiy et al., 2014; Doersch et al., 2015; Noroozi & Favaro, 2016). Recently, contrastive learning that uses different views of multiple augmented images has been an effective tool to learn rich representation from unsupervised data (Oord et al., 2018; Chen et al., 2020b; He et al., 2020; Tian et al., 2020), as these methods achieve comparable performance to fully-supervised models. While these works focus on improving standard generalization, we leverage representation learning, as proposed by Grill et al. (2020), to improve adversarial generalization.

**Semi- and self-supervised learning for adversarial robustness.** Schmidt et al. (2018) showed that learning adversarially robust models requires more data. As such, adversarial robustness with unlabeled data has recently drawn a lot of attention. We highlight the works by Uesato et al. (2019); Carmon et al. (2019); Zhai et al. (2019a) which leveraged labeled data to train a standard classifier that is in turn used to *pseudo-label* the remaining unlabeled data. However, as shown by Uesato et al. (2019); Carmon et al. (2019); Zhai et al. (2019a), when only 10% of the CIFAR-10 labels are available the robust accuracy drops significantly. In this paper, we focus on improving adversarial robustness in the low-label regime by leveraging unlabeled data (e.g., when 1%–10% of labels are available) to build robust representations. The result is a technique that significantly outperforms state-of-the-art *pseudo-labeling* techniques in the low-label regime and remains competitive with adversarial training when all labels are available. Chen et al. (2020a) also study adversarial pre-training on self-supervised tasks and demonstrate that they can train robust representations. However, to the contrary of our approach, their method does not preserve the robustness of their resulting representations on downstream tasks (unless robust fine-tuning is used). Hendrycks et al. (2019) combine supervised adversarial training with an additional self-supervised head. They demonstrate that they can improve on standard adversarial training, but do not learn general representations. We also highlight the recent work by Kim et al. (2020) which combines adversarial training with contrastive learning. Our method reaches comparable robust accuracy, but is more scalable (as it is not based on contrastive learning which requires large batch sizes). To the contrary of Kim et al., we also study the transferability of robust representations and focus on the low-label regime.

## 3 METHOD

In this section, we explain BYORL which elegantly combines adversarial training with BYOL. Hence, we start by giving a brief description of adversarial training and BYOL.

### 3.1 ADVERSARIAL TRAINING

Madry et al. (2017) formulate a saddle point problem whose goal is to find model parameters $\boldsymbol{\theta}$ that minimize the adversarial risk:

$$\mathbb{E}_{(\boldsymbol{x},y)\sim\mathcal{D}}\left[\max_{\boldsymbol{\delta}\in\mathbb{S}}l(f(\boldsymbol{x}+\boldsymbol{\delta};\boldsymbol{\theta}),y)\right] \tag{1}$$

where $\mathcal{D}$ is a data distribution over pairs of examples $\boldsymbol{x}$ and corresponding labels $y$, $f(\cdot;\boldsymbol{\theta})$ is a model parametrized by $\boldsymbol{\theta}$, $l$ is a suitable loss function (such as the $0-1$ loss in the context of classification tasks), and $\mathbb{S}$ defines the set of allowed perturbations (i.e., the adversarial input set or threat model).

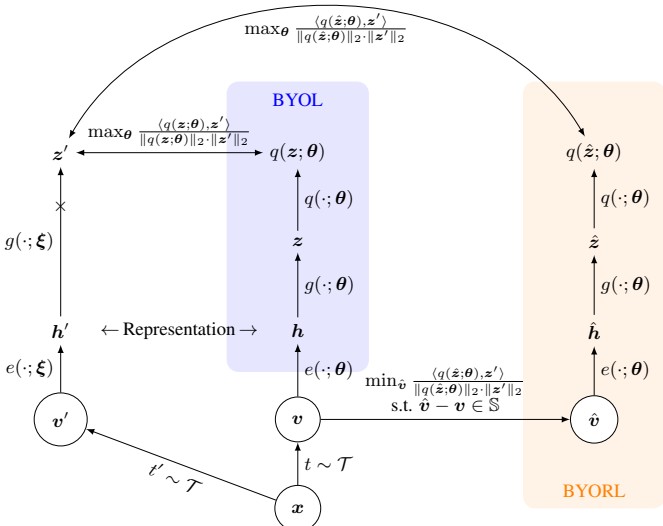

Figure 2: Flow diagram that highlights the difference between BYOL and BYORL. Whereas BYOL directly tries to maximize the cosine similarity between $q(\boldsymbol{z};\boldsymbol{\theta})$ and $\boldsymbol{z}'$, BYORL first executes an adversarial attack to retrieve an alternative image $\hat{\boldsymbol{v}}$.

Several methods (also known as "attacks") have been proposed to find adversarial examples (and effectively solve the inner maximization problem in Eq. 1). Classical adversarial training as proposed by Madry et al. (2017) uses Projected Gradient Descent (PGD),[2] which replaces the impractical $0-1$ loss $l$ with the cross-entropy loss $\hat{l}$ and computes an adversarial perturbation $\hat{\boldsymbol{\delta}} = \boldsymbol{\delta}^{(K)}$ in $K$ gradient ascent steps of size $\alpha$ as

$$\boldsymbol{\delta}^{(k+1)} \leftarrow \text{proj}_{\mathbb{S}}\left(\boldsymbol{\delta}^{(k)} + \alpha\nabla_{\boldsymbol{\delta}^{(t)}}\hat{l}(f(\boldsymbol{x} + \boldsymbol{\delta}^{(k)};\boldsymbol{\theta}),y)\right) \tag{2}$$

where $\boldsymbol{\delta}^{(0)}$ is chosen at random within $\mathbb{S}$, and where $\text{proj}_{\mathbb{A}}(\boldsymbol{a})$ projects a point $\boldsymbol{a}$ back onto a set $\mathbb{A}$. Finally, for each example $\boldsymbol{x}$ with label $y$, adversarial training minimizes the loss given by

$$\mathcal{L}_{\boldsymbol{\theta}}^{\text{AT}} = \hat{l}(f(\boldsymbol{x} + \hat{\boldsymbol{\delta}};\boldsymbol{\theta}),y) \approx \max_{\boldsymbol{\delta}\in\mathbb{S}}\hat{l}(f(\boldsymbol{x} + \boldsymbol{\delta};\boldsymbol{\theta}),y) \tag{3}$$

where $\hat{\boldsymbol{\delta}}$ is given by Eq. 2 and $\hat{l}$ is the softmax cross-entropy loss.

### 3.2 BOOTSTRAP YOUR OWN LATENTS

Many successful self-supervised learning approaches learn image representations by identifying whether different views belong to the same image (Dosovitskiy et al., 2014; Wu et al., 2018). Whereas contrastive methods formulate this prediction problem into one of discrimination (i.e., from the representation of an augmented view, they learn to discriminate between the representation of another augmented view of the same image, and the representations of augmented views of other images), BYOL relies on two neural networks: an online and a target network, that interact and learn from each other. The goal of the online network is to predict the target network representation of the same image under different augmented views, where the target network itself is defined by an exponential moving average of the online network parameters. We selected BYOL as the basis of our proposed method not only because it is currently the most successful representation learning technique, but also because it is more amenable to adversarial training, to contrary of contrastive methods which require large batch sizes (Chen et al., 2020b; Oord et al., 2018) or memory banks He et al. (2020).

As shown in Fig. 2, the online network is composed of three stages: an encoder $e(\cdot;\boldsymbol{\theta})$, a projector $g(\cdot;\boldsymbol{\theta})$ and a predictor $q(\cdot;\boldsymbol{\theta})$. Omitting the predictor, the target network has the same architecture

---

[2]There exists a few variants of PGD which normalize the gradient step differently (e.g., using its sign or $\ell_2$-norm depending on the threat model).

as the online network, but uses a different set of weights $\boldsymbol{\xi}$. As explained by Grill et al. (2020), in order to enhance representations while preventing their collapse, the target network's weights are allowed to change slowly throughout training. More precisely, given a decay rate $\tau \in [0, 1]$, after each training step, the parameters $\boldsymbol{\xi}$ are updated as $\boldsymbol{\xi} \leftarrow \tau\boldsymbol{\xi} + (1 - \tau)\boldsymbol{\theta}$. Given an image $\boldsymbol{x}$, and two augmentations $t, t' \sim \mathcal{T}$ sampled from a set of augmentations (e.g., random crops or recolorizations), BYOL produces two augmented views $\boldsymbol{v} = t(\boldsymbol{x})$ and $\boldsymbol{v}' = t'(\boldsymbol{x})$. The first view passes through the online network, producing a representation $\boldsymbol{h} = e(\boldsymbol{x}; \boldsymbol{\theta})$ and a projection $\boldsymbol{z} = g(\boldsymbol{h}; \boldsymbol{\theta})$. The second view similarly passes through the target network, producing a target projection $\boldsymbol{z}' = g \circ e(\boldsymbol{v}'; \boldsymbol{\xi})$. Finally, given an online prediction $q(\boldsymbol{z}; \boldsymbol{\theta})$ (which should be predictive of the target projection), BYOL minimizes the loss

$$\mathcal{L}_{\boldsymbol{\theta}}^{\text{BYOL}} = \left\| \frac{q(\boldsymbol{z}; \boldsymbol{\theta})}{\|q(\boldsymbol{z}; \boldsymbol{\theta})\|_2} - \frac{\boldsymbol{z}'}{\|\boldsymbol{z}'\|_2} \right\|_2^2 = 2 - 2 \cdot \frac{\langle q(\boldsymbol{z}; \boldsymbol{\theta}), \boldsymbol{z}' \rangle}{\|q(\boldsymbol{z}; \boldsymbol{\theta})\|_2 \cdot \|\boldsymbol{z}'\|_2}. \tag{4}$$

At the end of training, everything but $e$ and $\boldsymbol{\theta}$ is discarded and only the representation $e(\boldsymbol{x}; \boldsymbol{\theta})$ of an image $\boldsymbol{x}$ is used by downstream applications.

### 3.3 BOOTSTRAP YOUR OWN ROBUST LATENTS

We now introduce an effective and elegant approach to learn adversarially robust representations. Whereas previous (semi-)supervised techniques that are directly based on adversarial training require the presence of labels to produce adversarially robust neural networks (from which robust representations can be extracted at intermediate layers), our method can operate without any labels. As shown in the experimental section, the resulting representations can then be used to train linear classifiers (on top of these representation) that are intrinsically robust to adversaries. Our method, named Bootstrap Your Own Robust Latents or BYORL, consists of combining BYOL with adversarial training. A diagram summarizing BYORL is visible in Fig. 2.

For conciseness, we will denote by $\gamma = g \circ e$ the composition of the encoder and projector and by $\kappa = q \circ g \circ e$ the composition of the encoder, projector and predictor. Similarly to BYOL, BYORL starts by generating two views $\boldsymbol{v} = t(\boldsymbol{x})$ and $\boldsymbol{v}' = t'(\boldsymbol{x})$ of the same image $\boldsymbol{x}$. While the second view goes through the target network unmodified to produce a target projection $\boldsymbol{z}' = \gamma(\boldsymbol{v}'; \boldsymbol{\xi})$, the first view is further augmented via an adversarial attack. The goal of the adversarial attack is to maximize the disagreement between the online and target networks while respecting the threat model described by $\mathbb{S}$ (see subsection 3.1). To this end, we would like to find an optimal perturbation $\boldsymbol{\delta}^\star \in \mathbb{S}$ that minimizes the resulting cosine similarity between the online prediction $\kappa(\boldsymbol{v} + \boldsymbol{\delta}; \boldsymbol{\theta})$ and target projection $\boldsymbol{z}'$:

$$\boldsymbol{\delta}^\star = \arg\min_{\boldsymbol{\delta} \in \mathbb{S}} \frac{\langle \kappa(\boldsymbol{v} + \boldsymbol{\delta}; \boldsymbol{\theta}), \boldsymbol{z}' \rangle}{\|\kappa(\boldsymbol{v} + \boldsymbol{\delta}; \boldsymbol{\theta})\|_2 \cdot \|\boldsymbol{z}'\|_2}. \tag{5}$$

Like adversarial training, we can leverage PGD to approximate $\boldsymbol{\delta}^\star$ by $\hat{\boldsymbol{\delta}}$. Taking $K$ steps of size $\alpha$, resulting in $\hat{\boldsymbol{\delta}} = \boldsymbol{\delta}^{(K)}$, we have

$$\boldsymbol{\delta}^{(k+1)} \leftarrow \text{proj}_{\mathbb{S}} \left( \boldsymbol{\delta}^{(k)} + \alpha \nabla_{\boldsymbol{\delta}^{(t)}} \frac{\langle \kappa(\boldsymbol{v} + \boldsymbol{\delta}^{(t)}; \boldsymbol{\theta}), \boldsymbol{z}' \rangle}{\|\kappa(\boldsymbol{v} + \boldsymbol{\delta}^{(t)}; \boldsymbol{\theta})\|_2 \cdot \|\boldsymbol{z}'\|_2} \right) \tag{6}$$

where $\boldsymbol{\delta}^{(0)}$ is chosen at random within $\mathbb{S}$. Finally, we seek to maximize the agreement between the adversarially modified online prediction $\kappa(\boldsymbol{v} + \hat{\boldsymbol{\delta}}; \boldsymbol{\theta})$ and the target projection $\boldsymbol{z}'$ by updating the online weights $\boldsymbol{\theta}$ as to minimize the following loss:

$$\mathcal{L}_{\boldsymbol{\theta}}^{\text{BYORL}}(\boldsymbol{v}, \boldsymbol{v}') = 2 - 2 \cdot \frac{\langle \kappa(\boldsymbol{v} + \hat{\boldsymbol{\delta}}; \boldsymbol{\theta}), \gamma(\boldsymbol{v}'; \boldsymbol{\xi}) \rangle}{\|\kappa(\boldsymbol{v} + \hat{\boldsymbol{\delta}}; \boldsymbol{\theta})\|_2 \cdot \|\gamma(\boldsymbol{v}'; \boldsymbol{\xi})\|_2} \approx 2 - 2 \cdot \min_{\boldsymbol{\delta} \in \mathbb{S}} \frac{\langle \kappa(\boldsymbol{v} + \boldsymbol{\delta}; \boldsymbol{\theta}), \boldsymbol{z}' \rangle}{\|\kappa(\boldsymbol{v} + \boldsymbol{\delta}; \boldsymbol{\theta})\|_2 \cdot \|\boldsymbol{z}'\|_2}. \tag{7}$$

Here are a few additional considerations. First, we symmetrize the loss $\mathcal{L}_{\boldsymbol{\theta}}^{\text{BYORL}}$ in Eq. 7 by feeding $\boldsymbol{v}'$ to the online network and $\boldsymbol{v}$ to the target network. The adversarial attack is executed on $\boldsymbol{v}'$ instead of $\boldsymbol{v}$ and tries to minimize the cosine similarity between the online prediction $\kappa(\boldsymbol{v}' + \boldsymbol{\delta}; \boldsymbol{\theta})$ and target projection $\gamma(\boldsymbol{v}; \boldsymbol{\xi})$: $\mathcal{L}_{\boldsymbol{\theta}}^{\text{symmetric}}(\boldsymbol{v}, \boldsymbol{v}') = \mathcal{L}_{\boldsymbol{\theta}}^{\text{BYORL}}(\boldsymbol{v}, \boldsymbol{v}') + \mathcal{L}_{\boldsymbol{\theta}}^{\text{BYORL}}(\boldsymbol{v}', \boldsymbol{v})$. Second, one can observe that the adversarial attack is always performed through the online network. We could similarly perform the attack through the target network, but we found that training was less stable as batch

statistics (needed by batch normalization) were not representative of statistics induced by adversarial examples (as the online network would receive clean rather than adversarial images). Third, instead of the proposed method, we could imagine making two passes through the online network (for both the clean and adversarial images) and maximizing the agreement between both online predictions (in addition to maximizing the agreement with the target projection). We found that this increased the risk of representation collapse as this adds an incentive for the online network to output constant predictions (i.e., collapsed representations are the perfect defense against adversarial attacks).

## 4 EXPERIMENTS

We assess the performance of BYORL across multiple axes. We evaluate the robustness of the resulting representations by training robust linear classifiers on top of these representations. First, we compare to the performance of various classifiers (comparing BYORL with *pseudo-labeling*). Second, we study how these robust representations transfer to unseen new tasks. Finally, we also evaluate whether robustness transfers to downstream tasks – even when the final task is not treated as being adversarial.

### 4.1 SETUP AND IMPLEMENTATION DETAILS

We highlight here the most important components and defer some of the details to Appendix A.

**Architecture.** We use a convolutional residual network (He et al., 2015) with 34 layers (Pre-Activation ResNet-34) as our encoder $e$. We also use wider (from $\times 1$ to $\times 4$) ResNets. The projector $g$ and predictor $q$ networks are MLPs with hidden dimension $4096$ and output dimension $256$.

**Outer optimization.** We use the LARS optimizer (You et al., 2017) with a cosine learning rate schedule (Loshchilov & Hutter, 2017) over 1000 epochs. We set the learning rate to 2 and use a global weight decay parameter of $5 \cdot 10^{-4}$. For the target network, the exponential moving average parameter $\tau$ starts from 0.996 and is increased to one during training. We use a batch size of $512$.

**Inner optimization.** The inner minimization in Eq. 7 is implemented using $K$ PGD steps (constrained by an $\ell_2$ or $\ell_\infty$ norm-bounded ball). Unless specified otherwise, we set $K$ to 40 and use an adaptive step size $\alpha$ (see Algorithm 1 in Croce & Hein, 2020). For $\ell_\infty$ and $\ell_2$ norm-bounded perturbations, the gradients in Eq. 6 are first normalized to their sign or by their $\ell_2$ norm, respectively.

**Evaluation protocol.** We evaluate the performance of BYORL on CIFAR-10 against adversarial $\ell_2$ and $\ell_\infty$ norm-bounded perturbations (CIFAR-100 and IMAGENET results are in the appendix). For that purpose, we train a linear classifier parametrized by coefficients $\boldsymbol{W}$ and offsets $\boldsymbol{b}$ on top of frozen BYORL representations, following the procedure described in Kolesnikov et al. (2019); Chen et al. (2020b). The linear model is either trained in a non-robust manner (i.e., $\min_{\boldsymbol{W},\boldsymbol{b}} \mathbb{E}_{(\boldsymbol{x},y)\in\mathcal{D}}\hat{l}(\boldsymbol{W}e(\boldsymbol{x};\boldsymbol{\theta}) + \boldsymbol{b}, y))$ or adversarially (i.e., $\min_{\boldsymbol{W},\boldsymbol{b}} \mathbb{E}_{(\boldsymbol{x},y)\in\mathcal{D}} \max_{\boldsymbol{\delta}\in\mathbb{S}}\hat{l}(\boldsymbol{W}e(\boldsymbol{x} + \boldsymbol{\delta};\boldsymbol{\theta}) + \boldsymbol{b}, y))$. We then compute the robust accuracy, which is the accuracy of the combined model $\boldsymbol{W}e(\cdot;\boldsymbol{\theta}) + \boldsymbol{b}$ against adversarial attacks (i.e., we count a successful attack as a misclassification): $1 - \mathbb{E}_{(\boldsymbol{x},y)\in\mathcal{D}} \max_{\boldsymbol{\delta}\in\mathbb{S}} l(\boldsymbol{W}e(\boldsymbol{x}+\boldsymbol{\delta};\boldsymbol{\theta}) + \boldsymbol{b}, y)$. In order to get faithful results, all models are evaluated using a strong attack which combines elements of the AutoAttack procedure (Croce & Hein, 2020) with the MultiTargeted attack (Gowal et al., 2019). Namely, we use a sequence of AutoPGD on the cross-entropy loss with 5 restarts and 100 steps, AutoPGD on the difference of logits ratio loss with 5 restarts and 100 steps, MultiTargeted on the margin loss with 20 restarts and 200 steps and Square (Andriushchenko et al., 2019), an efficient black-box attack, with 5000 queries.

**Baseline.** Throughout the experimental section, we compare BYORL with adversarial training (combined with *pseudo-labeling* to handle missing labels). *Pseudo-labeling* is currently, to the best of our knowledge, the most successful semi-supervised method for learning adversarially robust models (Carmon et al., 2019; Uesato et al., 2019; Zhai et al., 2019a; Najafi et al., 2019). More specifically, we use Unsupervised Adversarial Training with Fixed Targets (UAT-FT) (Uesato et al., 2019). When 100% of the labels are available UAT-FT is equivalent to classical adversarial training, as proposed by Madry et al. (2017). In settings where less than 100% of the labels are available, we train a separate non-robust model (with an architecture identical to the robust model being trained) on the available labeled data and use it to *pseudo-label* the rest of the unlabeled images. UAT-FT uses the same network architectures than those used by BYORL.

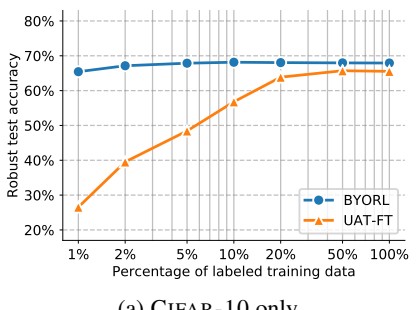 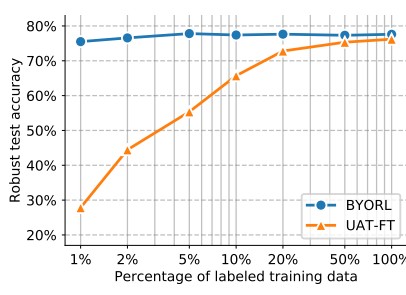

(a) CIFAR-10 only  (b) CIFAR-10 and 80M-TINYIMAGES

Figure 3: Accuracy under $\ell_2$ attack of size $\epsilon = 128/255$ for different CIFAR-10 models as a function of the ratio of available labels. Panel a restricts the available data to CIFAR-10 only (labeled and unlabeled), while panel b uses 500K additional unlabeled images extracted from 80M-TINYIMAGES.

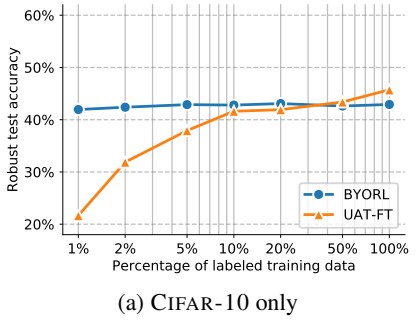 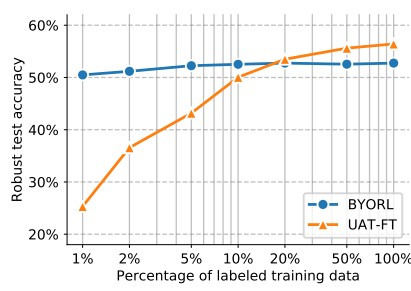

(a) CIFAR-10 only  (b) CIFAR-10 and 80M-TINYIMAGES

Figure 4: Accuracy under $\ell_\infty$ attack of size $\epsilon = 8/255$ for different CIFAR-10 models as a function of the ratio of available labels. Panel a restricts the available data to CIFAR-10 only (labeled and unlabeled), while panel b uses 500K additional unlabeled images extracted from 80M-TINYIMAGES.

## 4.2 RESULTS

**Robustness on CIFAR-10.** We evaluate BYORL and UAT-FT on a wide range of tasks across different threats for various amounts of available labels. As is typical in the literature (Rice et al., 2020; Augustin et al., 2020), we evaluate our models on CIFAR-10 against $\ell_2$ and $\ell_\infty$ norm-bounded perturbations of size $\epsilon = 128/255$ and $\epsilon = 8/255$ (CIFAR-100 and IMAGENET are evaluated in the appendix). CIFAR-10 contains 60K images (i.e., 50K in the train set and 10K in the test set). As such, when we evaluate on 1% of labeled data, we only use 500 random labeled images from CIFAR-10 (we do not artificially balance the number of labels per class). As done in Carmon et al. (2019) and Uesato et al. (2019), we also explore the limits of BYORL in the setting where additional unlabeled data is available. This additional data is extracted from 80M-TINYIMAGES and consists of 500K unlabeled $32 \times 32$ images[3]. In settings without this additional data, we use a ResNet-$34 \times 2$, whereas in settings with this additional data, we use a ResNet-$34 \times 4$.

Fig. 3 shows the robust accuracy of BYORL and UAT-FT on the full CIFAR-10 test set against $\ell_2$ norm-bounded perturbations (similar figures that show clean accuracy are available in Appendix B). We observe that linear classifiers trained on top of robust BYORL representations are more robust than those trained with UAT-FT. In particular, we highlight than when only 500 labeled images are available, BYORL remains competitive with state-of-the-art methods that use all labels: without additional data from 80M-TINYIMAGES, BYORL reaches 65.43% compared to 69.24% (Engstrom et al., 2019); with additional data from 80M-TINYIMAGES, BYORL reaches 75.50% compared to 72.91% (Augustin et al., 2020).

Fig. 4 shows the robust accuracy of BYORL and UAT-FT on the full CIFAR-10 test set against $\ell_\infty$ norm-bounded perturbations. Again, we can observe that BYORL remains competitive: in the low-label regime, BYORL surpasses UAT-FT by a significant margin (up to $2\times$ more accurate); in the high-label regime BYORL loses a few percentage points. Perhaps surprisingly, under both threat models (i.e., $\ell_\infty$ and $\ell_2$), BYORL reaches peak performance with 5% of the labels (and beyond).

---

[3]We use the dataset from Carmon et al. (2019) available at https://github.com/yaircarmon/semisup-adv.

Table 1: Robust accuracy (under adversarial attack) obtained by finetuning a linear head on top of robust representations trained on CIFAR-10.

| METHOD | NORM | RADIUS | STL-10 | | | CIFAR-100 | | |
|---|---|---|---|---|---|---|---|---|
| | | | 1% | 10% | 100% | 1% | 10% | 100% |
| BYORL | $\ell_2$ | $\epsilon = 128/255$ | 33.85% | **53.23%** | **57.88%** | **10.09%** | **22.51%** | **28.24%** |
| UAT-FT | | | **37.71%** | 42.16% | 53.23% | 3.68% | 10.80% | 14.43% |
| BYORL | $\ell_\infty$ | $\epsilon = 8/255$ | **24.18%** | 36.30% | 37.28% | **5.28%** | **10.12%** | **14.82%** |
| UAT-FT | | | 23.47% | **36.52%** | **37.79%** | 2.18% | 4.88% | 7.21% |

Table 2: Clean (no perturbations) and robust (under adversarial attack) accuracy obtained when training robust and non-robust representations on CIFAR-10 against $\ell_2$ norm-bounded perturbations of size $\epsilon = 128/255$. We evaluate the representations by finetuning a robust and non-robust linear head on CIFAR-10, STL-10 and CIFAR-100. Table 6 in the appendix shows $\ell_\infty$ perturbations.

| TRAINING OF | | NORM | RADIUS | CIFAR-10 | |
|---|---|---|---|---|---|
| REPRESENTATION | LINEAR HEAD | | | Clean | Robust |
| Robust (BYORL) | Robust (AT) on CIFAR-10 | | | 93.01% | **77.61%** |
| Robust (BYORL) | Non-robust on CIFAR-10 | $\ell_2$ | $\epsilon = 128/255$ | 93.19% | 77.09% |
| Non-robust (BYOL) | Robust (AT) on CIFAR-10 | | | 91.23% | 0.05% |
| Non-robust (BYOL) | Non-robust on CIFAR-10 | | | **94.76%** | 0.00% |
| FINETUNING OF LINEAR HEAD | | | | STL-10 | |
| Robust (BYORL) | Robust (AT) on STL-10 | $\ell_2$ | $\epsilon = 128/255$ | 76.49% | **57.88%** |
| Robust (BYORL) | Non-robust on STL-10 | | | **77.54%** | 57.66% |
| FINETUNING OF LINEAR HEAD | | | | CIFAR-100 | |
| Robust (BYORL) | Robust (AT) on CIFAR-100 | $\ell_2$ | $\epsilon = 128/255$ | 48.34% | **28.24%** |
| Robust (BYORL) | Non-robust on CIFAR-100 | | | **49.20%** | 27.17% |

**Transfer to unseen tasks.** We evaluate our robust representations on other classification datasets to assess whether the features learned on CIFAR-10 are generic and thus useful across image domains, or if they are CIFAR-10-specific. As a comparison, we test whether pre-logits activations resulting from training a model using UAT-FT can also be used for transfer learning. For both representations, we train a robust linear model using adversarial training (see subsection 3.1) with different label availability on STL-10 and CIFAR-100 against $\ell_\infty$ and $\ell_2$ norm-bounded perturbations. Table 1 shows that BYORL representations result in equivalent or more robust models than UAT-FT representations. We note, however, that – at least on CIFAR-100 – the robust accuracy remains significantly lower than models trained directly on CIFAR-100.

**Transfer without adversarial training.** So far, the linear classifiers trained on top of BYORL representations were trained robustly using adversarial training. We now evaluate whether adversarial training is needed for downstream tasks. Conversely, we also verify that learning robust representations is needed to obtain robust linear classifiers. Table 2 shows the robust accuracy of four models: *(i)* an adversarially trained linear model on top of robust BYORL representations, *(ii)* a classically trained (not necessarily robust) linear model on top of robust BYORL representations, *(iii)* an adversarially trained linear model on top of non-robust BYOL representations, and *(iv)* a classically trained linear model on top of non-robust BYOL representations. Although not a guarantee in theory (Allen-Zhu & Li, 2020), we observe that the adversarial training of the linear classifier is, in practice, not necessary and that it is enough to train robust representations to obtain robust classifiers. Indeed, for all three considered downstream tasks (CIFAR-10, STL-10 and CIFAR-100), the resulting non-robustly trained linear classifiers are within a few percentage points of the robustly trained ones (similar results for IMAGENET are available in the appendix in Table 5).

## 5 CONCLUSION

In this work, we present BYORL, a modification of BYOL that enables us to train robust image representations. To the contrary of previous methods, BYORL does not require the presence of label information. In fact, it is even possible to use these robust representations to train adversarially robust classifiers on multiple downstream tasks (without the need to use adversarial training). Interestingly, classifiers using BYORL representations can be trained with as little as 500 labeled examples. Across all experiments, BYORL with 1% of labels (i.e., 500 labeled examples) matches or surpasses the performance of *pseudo-labeling* (implemented through UAT-FT) with 10-20% of labels (i.e., between 5K and 10K labeled examples).

ACKNOWLEDGMENTS

We would like to thank Jean-Baptiste Alayrac, Olivier Hénaff, Jean-Bastien Grill, Florian Strub and Florent Altché for helpful discussions throughout this work.

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

# A    EXPERIMENTAL SETUP

**Image augmentations.**    BYORL uses similar image augmentations to SimCLR (Chen et al., 2020b) and BYOL (Grill et al., 2020). First, a random patch of the image is selected and resized to $32 \times 32$ with a random horizontal flip, followed by a color distortion, consisting of a random sequence of brightness, contrast, saturation, hue adjustments, and an optional grayscale conversion. Because of the low image resolution of CIFAR-10 and CIFAR-100, we do not apply a final Gaussian blur and solarization to the patches.

**Architecture.**    For consistency with prior work on adversarial robustness (Rice et al., 2020; Wong et al., 2020), we use a convolutional residual network (He et al., 2015) with 34 layers (Pre-Activation ResNet-34) as our encoder $e$. We also use wider (from $\times 1$ to $\times 4$) ResNets. The final representation $\boldsymbol{h}$ is the output of the final average pooling layer, which has a feature dimension of $2048$ (when the width multiplier is $\times 1$). The projector $g$ and predictor $q$ networks are *multi-layer perceptrons*. They consist of a linear layer with output size $4096$ followed by batch normalization (Ioffe & Szegedy, 2015), rectified linear units (ReLU) (Nair & Hinton, 2010), and a final linear layer with output dimension $256$.

**Outer optimization.**    We use the LARS optimizer (You et al., 2017) with a cosine decay learning rate schedule (Loshchilov & Hutter, 2017) over $1000$ epochs, with a warm-up period of $10$ epochs. We set the learning rate to $2$ and use a global weight decay parameter of $5 \cdot 10^{-4}$. For the target network, the exponential moving average parameter $\tau$ starts from $0.996$ and is increased to one during training using a cosine schedule. We use a batch size of $512$ split over $32$ Google Cloud TPU v3 cores. We train linear classifiers using the Adam optimizer (Kingma & Ba, 2014) for $50$ epochs with an initial learning of $2 \cdot 10^{-3}$ (the learning rate is decayed by a factor $10\times$ after $25$ epochs). For linear classifiers, we perform early stopping as suggested by Rice et al. (2020) using a separate set of $1024$ validation images (we do the same when training UAT-FT models). Note that linear models trained on frozen BYORL representations do not really need early stopping (as the linear model do not overfit significantly). However, UAT-FT and adversarial training are prone to robust overfitting and obtain significantly higher robust accuracy using early stopping.

**Inner optimization.**    The inner minimization in Eq. 7 is implemented using $K$ PGD steps (constrained by an $\ell_2$ or $\ell_\infty$ norm-bounded ball). In particular, unless specified otherwise, we set $K$ to $40$ and use an adaptive step size (as specified in Algorithm 1 in Croce & Hein, 2020). For $\ell_\infty$ and $\ell_2$ norm-bounded perturbations, the gradients in Eq. 6 are first normalized to their sign or by their $\ell_2$ norm, respectively. With this setup, training takes approximately $19$ hours for a ResNet-34 $\times$ 2 and $3$ days for a ResNet-34 $\times$ 4.

**Evaluation protocol.**    We evaluate the performance of BYORL on CIFAR-10 and CIFAR-100 against adversarial $\ell_2$ and $\ell_\infty$ norm-bounded perturbations. For that purpose, we train a linear classifier parametrized by coefficients $\boldsymbol{W}$ and offsets $\boldsymbol{b}$ on top of frozen BYORL representations, following the procedure described in Kolesnikov et al. (2019); Kornblith et al. (2019); Zhang et al. (2016); Chen et al. (2020b). The linear model is either trained in a non-robust manner (i.e., $\min_{\boldsymbol{W},\boldsymbol{b}} \mathbb{E}_{(\boldsymbol{x},y) \in \mathcal{D}} \hat{l}(\boldsymbol{W} e(\boldsymbol{x}; \boldsymbol{\theta}) + \boldsymbol{b}, y))$ or adversarially (i.e., $\min_{\boldsymbol{W},\boldsymbol{b}} \mathbb{E}_{(\boldsymbol{x},y) \in \mathcal{D}} \max_{\boldsymbol{\delta} \in \mathbb{S}} \hat{l}(\boldsymbol{W} e(\boldsymbol{x} + \boldsymbol{\delta}; \boldsymbol{\theta}) + \boldsymbol{b}, y))$. We then freeze the linear model and compute the accuracy of the combined model $\boldsymbol{W} e(\boldsymbol{x}; \boldsymbol{\theta}) + \boldsymbol{b}$ against adversarial attacks (i.e., we count a successful attack as a misclassification). In other words, we approximate the result of Eq. 1 using an inner-maximization procedure (i.e., adversarial attack). In order to get faithful results, all models are evaluated using a strong attack which combines elements of the AutoAttack procedure (Croce & Hein, 2020) with the MultiTargeted attack (Gowal et al., 2019). Namely, we use a sequence of AutoPGD on the cross-entropy loss with 5 restarts and 100 steps, AutoPGD on the difference of logits ratio loss with 5 restarts and 100 steps, MultiTargeted on the margin loss with 20 restarts and 200 steps and Square (Andriushchenko et al., 2019), an efficient black-box attack, with 5000 queries.

## B  ADDITIONAL EXPERIMENTS

In this section, we perform ablation studies on network architectures, strength of the attack used during representation learning, and color augmentation strength.

**Network size.**  In Table 3, we study the impact of model architectures on CIFAR-10 against $\ell_\infty$ norm-bounded perturbations of size $\epsilon = 8/255$. We train robust representations using wider ResNet, as in (Chen et al., 2020b; Grill et al., 2020), with 34 layers and a width multiplier 1, 2 and 4, and denoted by ResNet34×1, ResNet34×2, ResNet34×4. We observe that models increase performance in both clean and robust accuracy when model size increases. Without using additional unlabeled data, the robust accuracy saturates when using ResNet34×2. Hence, unless mentioned explicitly, in this paper, we use ResNet34×2 for settings without additional data.

Table 3: Clean (no perturbations) and robust (under adversarial attack) accuracy obtained by networks of different sizes on CIFAR-10 against $\ell_\infty$ norm-bounded perturbations of size $\epsilon = 8/255$.

| | | | CIFAR-10 | |
|---|---|---|---|---|
| ARCHITECTURE | NORM | RADIUS | Clean | Robust |
| RESNET 34x1 | | | 78.02% | 41.30% |
| RESNET 34x2 | $\ell_\infty$ | $\epsilon = 8/255$ | 80.80% | 43.24% |
| RESNET 34x4 | | | 81.32% | 43.19% |

**Attack strength.**  Table 4 shows results of training robust representations using different attack strengths on CIFAR-10 against $\ell_\infty$ norm-bounded perturbations of size $\epsilon = 8/255$. We study the effect of using standard PGD, as well as its adaptive variant named AutoPGD (Croce & Hein, 2020), with different step sizes using ResNet34×2 models. We observe that more robust representations are obtained as the attack strength increases. Although this finding is similar to previous observations (Qin et al., 2019), we notice that robust representation learning is more prone to gradient masking (resulting from using weak attacks during training) and requires stronger attacks then typically used in classical adversarial training. Throughout the paper, unless mentioned explicitly, we used AutoPGD with 40 steps.

Table 4: Clean (no perturbations) and robust (under adversarial attack) accuracy obtained by representations trained against attacks of different strength on CIFAR-10 against $\ell_\infty$ norm-bounded perturbations of size $\epsilon = 8/255$.

| | | | | CIFAR-10 | |
|---|---|---|---|---|---|
| ATTACK | STEPS | NORM | RADIUS | Clean | Robust |
| PGD | 5 | | | 85.26% | 1.05% |
| | 10 | $\ell_\infty$ | $\epsilon = 8/255$ | 81.44% | 37.88% |
| | 20 | | | 80.29% | 42.58% |
| | 40 | | | 79.60% | 43.41% |
| AUTOPGD | 10 | | | 80.49% | 40.59% |
| | 20 | $\ell_\infty$ | $\epsilon = 8/255$ | 80.53% | 43.27% |
| | 40 | | | 80.47% | 43.64% |
| | 60 | | | 80.69% | 43.64% |

**Impact of color augmentations.**  In Chen et al. (2020b) and Grill et al. (2020), the authors demonstrate the importance of using color augmentations, which is composed color jittering and color dropping. We follow the same setup as Chen et al. (2020b) and Grill et al. (2020), in Fig. 5, we perform an ablation study on the strength of color augmentations. We use AutoPGD with 20 steps on a ResNet34×2 model. We observe that BYORL is relatively stable in both clean and robust accuracy for color augmentation strength between 0.1 and 0.5. In this paper, we had fixed the color augmentation strength to be 0.5 across all the experiments. A better choice in retrospect would have been 0.3 which provides an improvement of 0.34%.

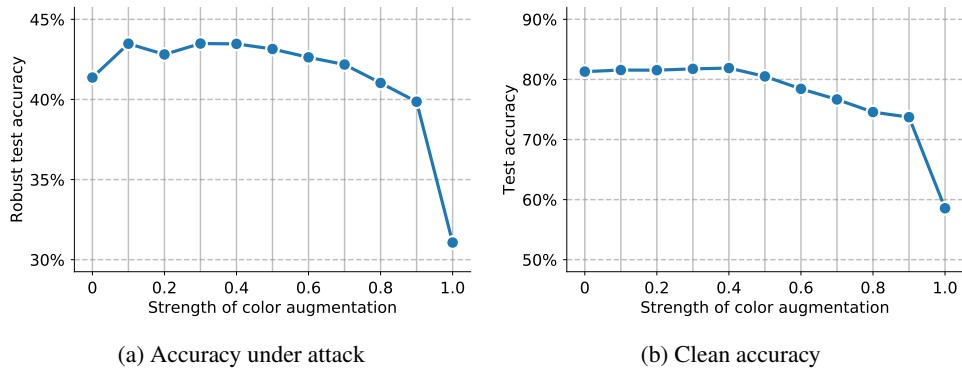

(a) Accuracy under attack          (b) Clean accuracy

Figure 5: Accuracy on CIFAR-10 for different strengths of the color augmentation. Panel a shows the accuracy under $\ell_\infty$ attacks of size $\epsilon = 8/255$, while panel b shows the corresponding clean accuracy.

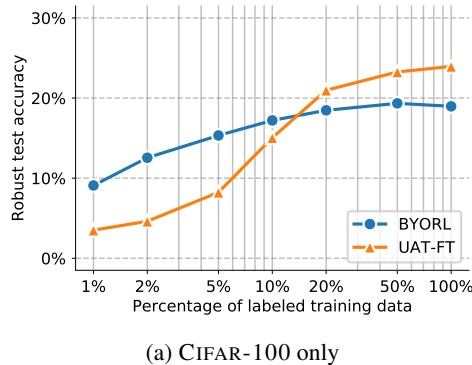

(a) CIFAR-100 only

Figure 6: Accuracy under $\ell_\infty$ attack of size $\epsilon = 8/255$ for different CIFAR-100 models as a function of the ratio of available labels. The available data is restricted to CIFAR-100 only (labeled and unlabeled).

**CIFAR-100.** We repeat the experiments in subsection 4.2 on CIFAR-100. For CIFAR-100, we expected BYORL to perform significantly better than UAT-FT as the number of classes is ten times larger than on CIFAR-10. However, Fig. 6 shows a more nuanced story and remains similar to Fig. 4. It is interesting to observe that BYORL tends to struggle with $\ell_\infty$ norm-bounded perturbations of size $\epsilon = 8/255$. As it has long been known that $\ell_\infty$ norm-bounded perturbations of size $\epsilon = 8/255$ are harder to cope with than $\ell_2$ norm-bounded perturbations of size $\epsilon = 128/255$, we posit that, at equal model capacity, the pretext task (consisting of augmenting two different views of the same image) needs to be slightly modified to accommodate the more difficult threat. Indeed, the investigation on the impact of the color augmentation strength showed that it is possible to improve robust representations by reducing the strength of these augmentations.

**IMAGENET.** We repeat a subset of the experiments from subsection 4.2 on IMAGENET. For IMAGENET, we train BYORL representations with a ResNet 50×4 for 300 epochs. Following Grill et al. (2020) and Chen et al. (2020b), we add random blur to the image preprocessing pipeline. We also change the learning rate from 2 to 1.2 and the initial target network weight decay rate $\tau$ to 0.999. We use a batch size of 1024 split over 128 Google Cloud TPU v3 cores. All other parameters remain identical to other experiments.

Table 5 shows that BYORL is capable of learning robust representations for larger datasets such as IMAGENET. In particular, our best model achieves 45.44% robust accuracy against $\ell_\infty$ norm-bounded perturbations of size $\epsilon = 4/255$ (as evaluated by the combined set of attacks consisting of AutoPGD on the cross-entropy loss with 5 restarts and 100 steps, AutoPGD on the difference of logits ratio loss with 5 restarts and 100 steps). Note that classical adversarial training achieves 39.7% (when evaluated against PGD on the cross-entropy loss with 1 restart and 100 steps), while the current state-of-the-art

is 47.00% (Qin et al., 2019). While BYORL does not reach the state-of-the-art in robustness, it is remarkable to see it perform so well with only 1% of available labels and reach 31.57% robust accuracy (e.g., classical non-robust supervised finetuning achieves 25.4% top-1 accuracy when no adversarial perturbations are present as stated in Zhai et al., 2019b). Finally, these results also confirm the results from Table 2 whereby training a non-robust linear classifier still achieves significant robust accuracy.

Table 5: Clean (no perturbations) and robust (under adversarial attack) accuracy obtained when training robust representations on IMAGENET against $\ell_\infty$ norm-bounded perturbations of size $\epsilon = 4/255$. We evaluate the representations by training/finetuning a robust and non-robust linear head on IMAGENET with varying numbers of labels. For completeness, we also add results for $\ell_2$ norm-bounded perturbations of size $\epsilon = 128/255$

| TRAINING OF LINEAR HEAD | NORM | RADIUS | IMAGENET (100%) Clean | Robust | IMAGENET (10%) Clean | Robust | IMAGENET (1%) Clean | Robust |
|---|---|---|---|---|---|---|---|---|
| Robust (AT) Non-robust | $\ell_\infty$ | $\epsilon = 4/255$ | 65.14% 65.27% | 45.44% 43.83% | 62.39% 62.40% | 41.58% 41.06% | 47.07% 47.64% | 31.57% 31.99% |
| Robust (AT) | $\ell_2$ | $\epsilon = 128/255$ | 69.64% | 65.48% | 66.39% | 61.90% | 55.06% | 51.00% |

**Transfer without adversarial training against $\ell_\infty$ norm-bounded perturbations.** In subsection 4.2, we evaluate how robust accuracy degrades when linear classifiers trained on top of BYORL representations are not trained robustly. In Table 2, we evaluate models trained against $\ell_2$ norm-bounded perturbations. In Table 6, we evaluate models trained against $\ell_\infty$ norm-bounded perturbations. In both cases, non-robust downstream classifiers are able to exhibit non-trivial levels of robustness that remain within a few percentage points of the robustly trained ones.

Table 6: Clean (no perturbations) and robust (under adversarial attack) accuracy obtained when training robust and non-robust representations on CIFAR-10 against $\ell_\infty$ norm-bounded perturbations of size $\epsilon = 8/255$. We evaluate the representations by training/finetuning a robust and non-robust linear head on CIFAR-10, STL-10 and CIFAR-100.

| TRAINING OF REPRESENTATION | LINEAR HEAD | NORM | RADIUS | CIFAR-10 Clean | Robust |
|---|---|---|---|---|---|
| Robust (BYORL) Robust (BYORL) | Robust (AT) on CIFAR-10 Non-robust on CIFAR-10 | $\ell_\infty$ | $\epsilon = 8/255$ | 84.96% 86.57% | **52.75%** 50.94% |
| FINETUNING OF LINEAR HEAD | | | | STL-10 | |
| Robust (BYORL) Robust (BYORL) | Robust (AT) on STL-10 Non-robust on STL-10 | $\ell_\infty$ | $\epsilon = 8/255$ | 65.40% **65.75%** | **37.79%** 36.51% |
| FINETUNING OF LINEAR HEAD | | | | CIFAR-100 | |
| Robust (BYORL) Robust (BYORL) | Robust (AT) on CIFAR-100 Non-robust on CIFAR-100 | $\ell_\infty$ | $\epsilon = 8/255$ | 37.80% **39.51%** | **14.82%** 12.21% |

## C ADDITIONAL FIGURES

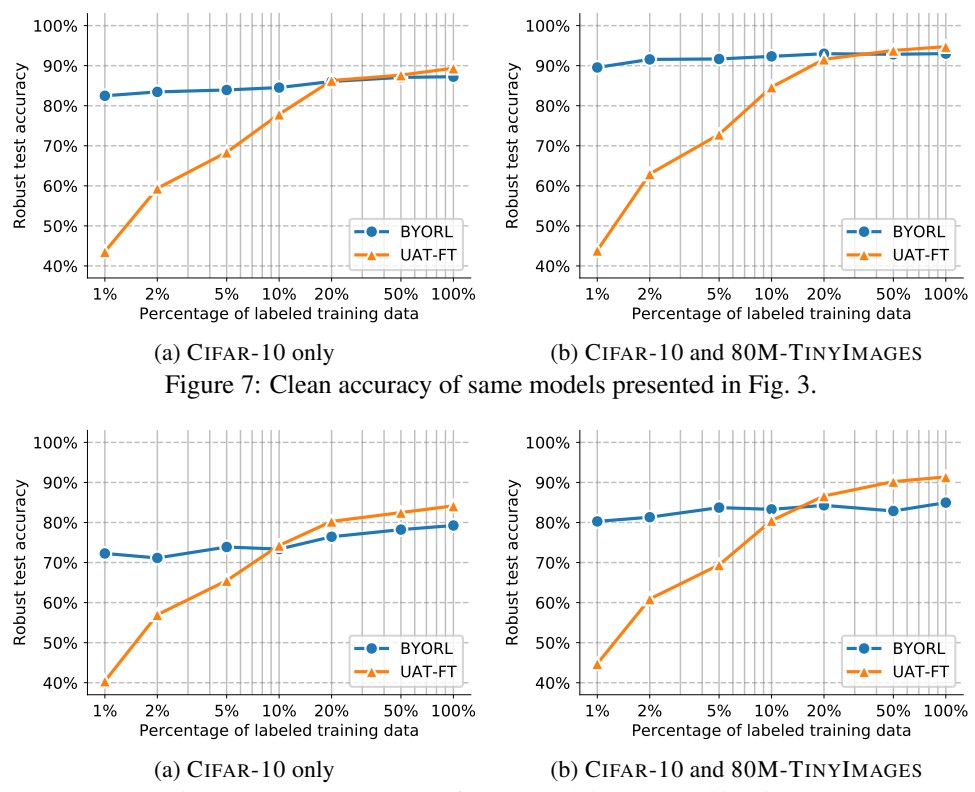



(a) CIFAR-10 only        (b) CIFAR-10 and 80M-TINYIMAGES

Figure 7: Clean accuracy of same models presented in Fig. 3.

(a) CIFAR-10 only        (b) CIFAR-10 and 80M-TINYIMAGES

Figure 8: Clean accuracy of same models presented in Fig. 4.



## D ANALYSIS OF RESULTING MODELS

In this section, we perform analysis on the adversarial loss landscape from the BYORL models in $\ell_2$ and $\ell_\infty$ cases to further examine whether adversarial accuracy is due to gradient masking (Uesato et al., 2018; Athalye et al., 2018). The analysis made here complements the use of the black-box Square (Andriushchenko et al., 2019) attack used by AutoAttack within our evaluation pipeline. For generating a loss landscape, we vary the input along a linear space defined by the worse perturbations found by PGD ($u$ direction) and a random Rademacher direction ($v$ direction). The $u$ and $v$ axes represent the magnitude of the perturbation added in each of these directions respectively and the $z$ axis represents the adversarial margin loss (Carlini & Wagner, 2017b).

$\ell_2$ **model.** In Fig. 9, we show the adversarial loss landscapes of 4 randomly selected CIFAR-10 test images from the BYORL model trained against $\ell_2$ perturbations on both CIFAR-10 and a subset of 80M-TINYIMAGES. The loss surfaces are generally smooth in Fig. 9, which provides evidence that the model performance is not due to gradient obfuscation. We also note that our rigorous evaluation already uses a black-box attack (i.e., Square Andriushchenko et al., 2019).

$\ell_\infty$ **model.** Fig. 10 shows the adversarial loss landscapes of 4 randomly selected CIFAR-10 test images from the BYORL model trained against $\ell_\infty$ perturbations on both CIFAR-10 and a subset of 80M-TINYIMAGES. We observe that the loss landscapes are generally smooth in Figure 10, which further suggests the model performance is not due to gradient obfuscation. We also note that our rigorous evaluation already uses a black-box attack (i.e., Square by Andriushchenko et al., 2019).

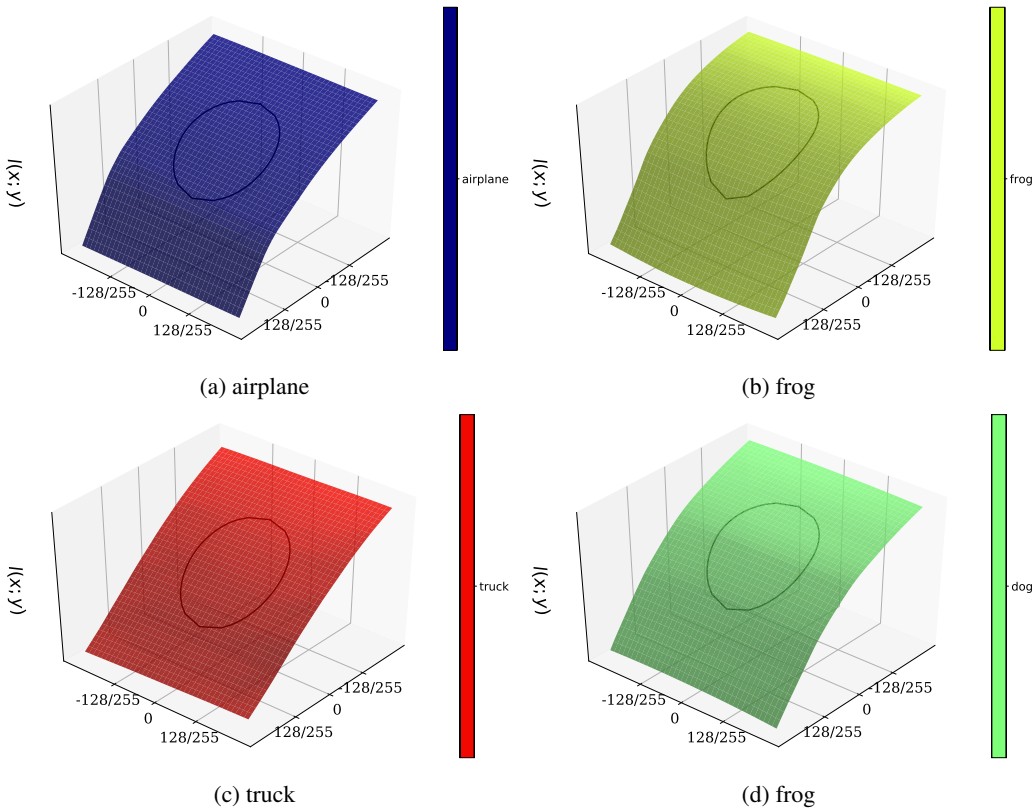

(a) airplane

(b) frog

(c) truck

(d) frog

Figure 9: Loss landscapes around different CIFAR-10 test images. It is generated by varying the input to the model, starting from the original input image toward either the worst attack found using PGD ($u$ direction) or a random Rademacher direction ($v$ direction). The loss used for these plots is the margin loss $z_y - \max_{i \neq y} z_i$ (i.e., a misclassification occurs when this value falls below zero). The model used is the BYORL model trained against $\ell_2$ perturbations on both CIFAR-10 and a subset of 80M-TINYIMAGES. The circular-shape represents the projected $\ell_2$ ball of size $\epsilon = 128/255$ around the nominal image.

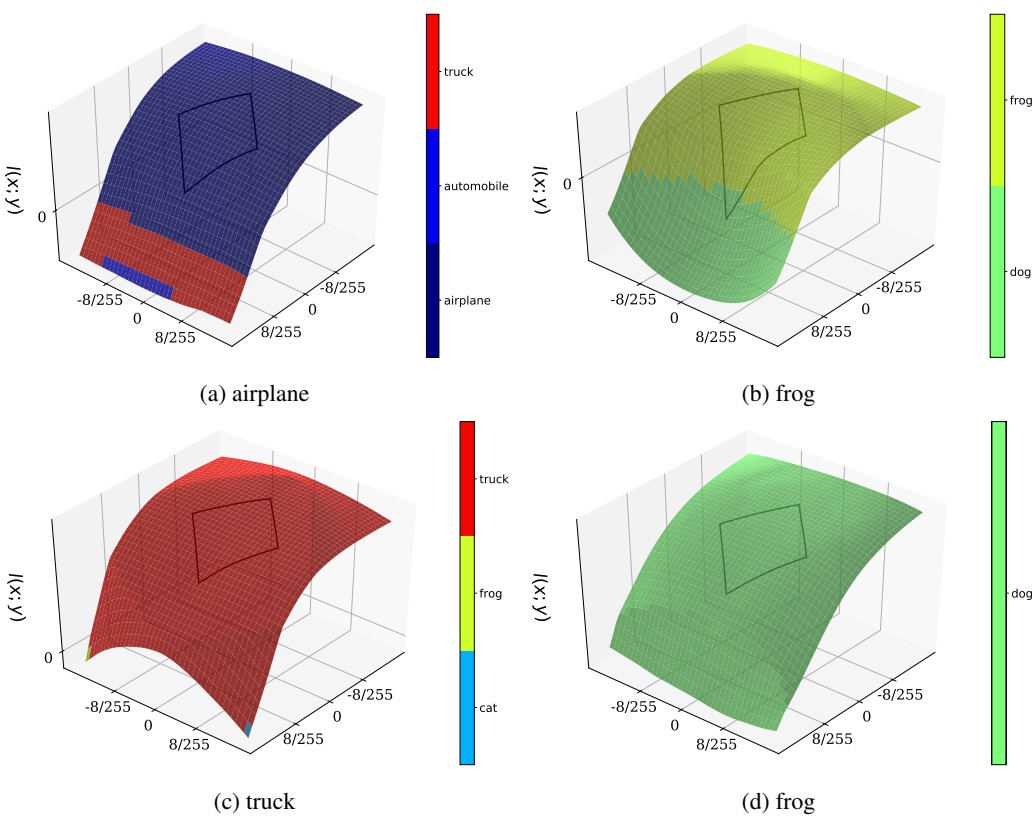

(a) airplane

(b) frog

(c) truck

(d) frog

Figure 10: Loss landscapes around the clean image of different CIFAR-10 test images. It is generated by varying the input to the model, starting from the original input image toward either the worst attack found using PGD ($u$ direction) or a random Rademacher direction ($v$ direction). The loss used for these plots is the margin loss $z_y - \max_{i \neq y} z_i$ (i.e., a misclassification occurs when this value falls below zero). The model used is the BYORL model trained against $\ell_\infty$ perturbations on both CIFAR-10 and a subset of 80M-TINYIMAGES. The diamond-shape represents the projected $\ell_\infty$ ball of size $\epsilon = 8/255$ around the nominal image.

