# OpenReview forum: "Self-supervised Adversarial Robustness for the Low-label, High-data Regime"
_ICLR.cc/2021/Conference — ICLR 2021 Poster_

### Official Review · AnonReviewer2 · 2020-10-17
**Review: SELF-SUPERVISED ADVERSARIAL ROBUSTNESS FOR THE LOW-LABEL, HIGH-DATA REGIME**

**Rating:** 7
**Confidence:** 4

**Review:**

##########################################################################
Summary:

The paper proposes a new self-supervised technique, Bootstrap Your Own Robust Latents (BYORL), based on an existing technique, BYOL. BYORL proposes to provide adversarially robust representations for low-label regimes. The paper claims that BYORL achieves state-of-the-art performance on CIFAR-10 even with data that is labeled as low as 1%. In fact, the authors highlight that the representations resulted from BYORL avoid the explicit training for adversarial robustness, because they are already robust.

##########################################################################

Reasons for score:

Overall, the paper in its current form is above the acceptance threshold. The proposed idea looks very encouraging. Provided the authors address some of the concerns the proposed method has a significant potential to the self-supervised adversarial robustness. If we look at the low-label regime alone, probably the paper has good contributions, that on ImageNet is not entirely convincing. If the authors convince me on that, happy to increase the score to clearly accept.
##########################################################################


Pros:

1. Overall, the paper is well written except for a few minor grammatical errors, easy to follow and understandable.

2. The paper discusses the existing literature and positions the proposed approach with respect to the state-of-the-art. The proposed method is definitely a decent contribution towards the field.

3. While robustness evaluations are good, the transfer to unseen and without adversarial training are much more encouraging.


Cons:

1. In abstract “... pioneered by four separate and simultaneous work in 2019,” should be “... pioneered by four separate and simultaneous works in 2019,”

2. “Since Madry et al. (2017), various modification to their ...” should be “Since Madry et al. (2017), various modifications to their ...”

3. As the % of labeled training data increases, there is not a significant improvement in robust test accuracy for CIFAR-10, that increases for CIFAR-100. Why is that happening for CIFAR-10?

4. On CIFAR-100, for anything over 10% labeled training data the other methods are still the state-of-the-art, the proposed method does not perform well. It again probably is due to the above observation of robust accuracy not increasing with a steep slope like the other methods.

5. What is robust accuracy, Zhang et al. 2019 or Uesato et al 2019 or others have it defined, nevertheless, please say in a sentence what it is.

6. On the ImageNet, the hypothesis of BYORL to get better as the labeled data increases is hard to buy given what we saw on CIFAR-10 and CIFAR-100. For those two, BYORL starts better and the state-of-the-art methods either reach BORL or outperform it. In ImageNet case, BYORL is outperformed with 1% itself, unless supported with empirical evidence, it is hard to believe the above statement of improvement.

7. Overall, the empirical results are satisfactory but not entirely convincing.

---

> ### Author Response · Authors · 2020-11-12
> **Clarifications and new ImageNet results**
>
> Thank you for the detailed review. We highlight that we obtained new results on ImageNet which now surpass the known state-of-the-art against l-inf perturbations of size 4/255 (see end of answer).
>
> > Overall, the paper is well written except for a few minor grammatical errors, easy to follow and understandable. The paper discusses the existing literature and positions the proposed approach with respect to the state-of-the-art. The proposed method is definitely a decent contribution towards the field. While robustness evaluations are good, the transfer to unseen and without adversarial training are much more encouraging.
>
> We appreciate that the main results of the paper were clear. Indeed, the fact that robust representations transfer well (even when using non-robust fine-tuning) is very encouraging and could impact many downstream applications.
>
> > In abstract “... pioneered by four separate and simultaneous work in 2019,” should be “... pioneered by four separate and simultaneous works in 2019,” + “Since Madry et al. (2017), various modification to their ...” should be “Since Madry et al. (2017), various modifications to their ...”
>
> Thank you. These are now corrected.
>
> > As the % of labeled training data increases, there is not a significant improvement in robust test accuracy for CIFAR-10, that increases for CIFAR-100. Why is that happening for CIFAR-10?
>
> A possible explanation could be that CIFAR-10 only contains 10 classes. Hence with 500 examples, the model would see 50 examples per class. For CIFAR-100, the model would only see 5 examples per class.
>
> > On CIFAR-100, for anything over 10% labeled training data the other methods are still the state-of-the-art, the proposed method does not perform well. It again probably is due to the above observation of robust accuracy not increasing with a steep slope like the other methods.
>
> The true reason why this happens may be two-fold: (i) CIFAR images have very low resolutions and self-supervised techniques rely of data augmentation schemes (such as heavy cropping) which might make things worse, especially when the number of classes increases (i.e., dataset diversity); (ii) BYORL has a much harder time with l-inf perturbations (at least for the hyper-parameters we chose). We also note that self-supervised techniques (with linear fine-tuning) do not yet match standard training performance in all situations (e.g., best supervised techniques reach > 88% top-1 accuracy on ImageNet, whereas the best self-supervised techniques reach 80% under linear evaluation), but they have the advantage of building general representations that can be used for many downstream tasks.
>
> > What is robust accuracy, Zhang et al. 2019 or Uesato et al 2019 or others have it defined, nevertheless, please say in a sentence what it is.
>
> This has been added to manuscript.
>
> > On the ImageNet, the hypothesis of BYORL to get better as the labeled data increases is hard to buy given what we saw on CIFAR-10 and CIFAR-100. For those two, BYORL starts better and the state-of-the-art methods either reach BORL or outperform it. In ImageNet case, BYORL is outperformed with 1% itself, unless supported with empirical evidence, it is hard to believe the above statement of improvement.
>
> Thankfully, we had anticipated this question and did perform new experiments on ImageNet in a bid to improve our original results.
>
> First, we highlight that Table 5 does not contain any results from the baseline (UAT), it only shows how robustness transfers (with robust and non-robust fine-tuning). Second, [1] showed that with 1% of labels, a standard classifier reaches 22% top-1 accuracy (we reach 32%), so it is anticipated that BYORL is better than UAT when only 1% of the labels are available.
>
> Most importantly, we were able to train a larger model (ResNet50-4x with lower color augmentation strength of 0.2 and an EMA decay rate of 0.998) and obtained a robust accuracy of 47.6% when using 100% of labels (against AutoPGD with 100 steps). This surpasses the best known supervised training result which is 47.0% (against PGD-100 and trained using a new regularizer) [2]. Standard adversarial training only reaches 39.7% with 100% of the labels (and UAT is equivalent to standard adversarial training when 100% of the labels are available). Hence, we are confident that overall BYORL will be better than UAT when only 10% or 1% of the labels are available (since it is better in the 100% setting). Note that full evaluations are still ongoing and we expect to update this thread and the paper early next week.
>
> > Overall, the empirical results are satisfactory but not entirely convincing.
>
> We hope that our answer (and the answers to other reviewers) provide some confidence in our results.
>
> [1] Chen et al., "A simple framework for contrastive learning of visual representations", 2020
> [2] Qin et al., "Adversarial Robustness through Local Linearization" , 2019

---

> > ### Author Response · Authors · 2020-11-16
> > **Updated ImageNet results**
> >
> > We have finished the evaluation of our newest ImageNet models (with robust fine-tuning) against eps = 4/255.
> > * With 1% of available labels: Robust accuracy is 28.59%, Clean accuracy is 44.64%.
> > * With 10% of available labels: Robust accuracy is 38.96%, Clean accuracy is 61.51%.
> > * With 100% of available labels: Robust accuracy is 41.83%, Clean accuracy is 63.29%.
> >
> > Overall, results are very encouraging as they surpass the performance of classical adversarial training [2]. We are now in the process of evaluating standard fine-tuning and will update the paper with these newer numbers.
> >
> > The previously reported number of 47.6% was computed against AutoPGD-100 with 1 restart, as opposed of the combination of AutoAttack and MultiTargeted which is used for the numbers above and throughout the paper.
> >
> > [2] Qin et al., "Adversarial Robustness through Local Linearization" , 2019

---

> > > ### Author Response · Authors · 2020-11-20
> > > **Non-robust finetuning results on ImageNet**
> > >
> > > The final results are now available in latest revision of manuscript. In particular, our models surpass classical adversarial training when 100% of the labels are available.

---

> > > > ### Author Response · Authors · 2020-11-23
> > > > **Thank you**
> > > >
> > > > Dear reviewer,
> > > >
> > > > As the rebuttal period finishes tomorrow, we would appreciate any additional comments or requests for clarification.
> > > >
> > > > Related to the following comment:
> > > >
> > > > > If we look at the low-label regime alone, probably the paper has good contributions, that on ImageNet is not entirely convincing. If the authors convince me on that, happy to increase the score to clearly accept.
> > > >
> > > > We highlight that the new results in the low-label regime (1% of labels) on ImageNet show a robust accuracy of 29.49% which surpasses by a large margin the clean accuracy obtained by a standard classifier for the same amount of labels. In the high-label regime (100% of labels), representations obtained by BYORL surpass classical adversarial training.

---

> > > > > ### Comment · AnonReviewer2 · 2020-11-24
> > > > > **Satisfactory and convincing results**
> > > > >
> > > > > The authors have addressed al most all of my concerns quite convincingly. In fact the large scale Imagenet results are a great addition to the proof of concept.
> > > > >
> > > > > Of course, increasing my earlier score to clear accept.

---

### Official Review · AnonReviewer3 · 2020-10-28
**New method, good improvments in low-label regime**

**Rating:** 7
**Confidence:** 4

**Review:**

This paper introduces a new algorithm for learning adversarially robust models in the semi-supervised setting, where a small amount of labeled data is available together with a sizeable unlabeled dataset. The proposed approach BYORL adapts an existing self-supervised learning method BYOL by introducing a new adversarial augmentation technique based on maximizing the cosine similarity between representations. BYORL is evaluated on CIFAR-10 and compared against a recent pseudo-labelling based approach UAT-FT for the semi-supervised setting, and is shown to outperform UAT-FT in terms of robust accuracy under $\ell_2$ and $\ell_\infty$ attacks under the low-labelled data regime. The representations learnt by BYORL are also shown to be better than that of UAT-FT when transferred to other datasets. Finally, robust representations are shown to be more important than learning a robust linear classifier on top.

Strengths:
- Significant improvements in robust accuracy in the low-label regime
- Interesting observations regarding transferability of robust representations and importance of final robust linear classifier
- Paper is generally clear and well-written

Weaknesses:
- RoCL as introduced by "Adversarial Self-Supervised Contrastive Learning", NIPS 2020 can also be applied to the semi-supervised setting, which somewhat compromises novelty. Ideally this method should be included in the experimental evaluations. This paper was cited but not discussed in the context of related work, but should be.
- Label budgets claimed do not include sizeable validation set used for early stopping, which can sometimes exceed the label budget (e.g. result with 500 labeled images also uses validation set of 1024 examples).
- Experimental evaluation could be strengthened in a few ways:
  * Clean accuracy comparison for Fig 3/4 - does UAT-FT do better in terms of clean accuracy?
  * It would be informative to include the robust accuracy of a model trained directly on STL-10 and CIFAR-100 in Table 2 to show how good the transferred representations are.
  * Experimental results seem to be from a single run; multiple runs would give an idea of the variance.

Overall, the robust accuracy improvements achieved in the low-label regime by BYORL are significant and the method is somewhat novel. The paper could be further strengthened by improving the experimental evaluation as described above. The highly related work RoCL should be discussed and ideally compared with.

Other comments:
- The paragraph after Eq 8 was confusing to me. What is meant by symmetrizing the loss? What is the final loss used to train the model after symmetrization? And why does the argument about batch-norm statistics not apply to the online network as well if the loss is symmetrized (which I took to mean that both models would be separately attacked and the losses added up)?
- Some analysis into the representations learnt would be helpful to give some insight into why the method works.
- Why does the method work better under $\ell_2$ attacks as compared to $\ell_\infty$ attacks?
- What is the effect of different transformations on robustness? Are the transformations important for robustness different from those important for clean classification?

*** Post Response Comments ***
I thank the authors for addressing the points raised. I am raising my score accordingly to 7.

Nit: The y-axis labels on Figure 7 and 8 should probably say "Clean accuracy" instead of "Robust test accuracy".

---

> ### Author Response · Authors · 2020-11-12
> **Clarifications and additional results**
>
> Thank you for the detailed review. We appreciate that the main results of the paper were clear. The adversarial low-label regime has not been widely studied and we believe to be among the first to show that, in practice, robust representations can transfer well (even when using standard finetuning).
>
> > RoCL [1] somewhat compromises novelty
>
> We have only made recently aware that [1] was accepted to NeurIPS. In fact, at submission time, we only had access to its ArXiv version. This being said:
> * BYORL is more scalable that RoCL as it is not based on contrastive learning which requires large batch sizes.
> * The results in [1] are conducted against a PGD-20 attack and, as such, results are not directly comparable. From a high-level view, partial finetuning and transfer results (using adversarial fine-tuning) seem on par (at least when comparing to the CIFAR-10 -> CIFAR-100 transfer against l-inf perturbations).
> * [1] neither studies the transferability of robust representations (when using standard finetuning), nor the low-label regime.
>
> We believe that both papers are complementary and have highlighted differences in the new revision of the manuscript.
>
> > Label budgets do not include validation set used for early stopping
>
> We used this additional validation set for fairness. In fact, the BYORL models do not need to be early stopped (i.e., best checkpoint is often the last checkpoint), but we wanted to give a fair chance to the baseline adversarial training method (which strongly benefits from early stopping) [2]. While 1024 may seem excessive, it is a only a constant offset (e.g., 1% of labels is really 3%, 50% of labels is really 52%), and both methods benefit from this additional data in the same way. We have added more information in the new paper revision.
>
> > Clean accuracy for Fig 3/4
>
> Apologies for not providing these numbers in the original submission (we have now added them to the appendix). Clean accuracy seems consistent (in terms of trend) with robust accuracy. In the low-label regime, BYORL does much better, while in the high-label regime UAT-FT does better.
>
> > informative to include the robust accuracy of a model trained directly on STL-10 and CIFAR-100 in Table 2
>
> We did not train models on STL-10 or CIFAR-100 directly. We note that this would be good to have and plan to add this in the near future (each evaluation against AutoAttack+MultiTargeted takes a few days to complete). We do, however, expect models trained directly on STL-10 and CIFAR-100 to be significantly better. As a point of comparison, models trained on CIFAR-100 using adversarial training reach 43.2% robust accuracy [2]. The main message from Table 2 is to demonstrate non-trivial robustness transfer (even when using standard fine-tuning instead of robust fine-tuning).
>
> > multiple runs would give an idea of the variance.
>
> We are conscious of this limitation. As each evaluation run takes a significant amount of time, we limited ourselves to a single training run. That being said, Fig. 3 and 4 can help estimate this variance. We observe that runs after 5% of labeled data obtain roughly the same robust accuracy (within 0.5% percentage points).
>
> > The paragraph after Eq 8 was confusing to me.
>
> We symmetrize the loss by swapping $v$ and $v'$ and summing both losses. The adversarial attack is always done through the online network. The updated manuscript now contains the complete symmetric loss (denoted $\mathcal{L}_\theta^\textrm{symmetric}$).
>
> > Some analysis into the representations learnt would be helpful
>
> We refrained from doing any additional analysis as there are no universally accepted method to analyze such representations. We could produce a t-SNE plot, but did not believe it would give any additional insight (beyond the ones we already have).
>
> > Why does the method work better under l2 attacks as compared to l-inf attacks?
>
> l2 attacks of size 128/255 are easier for models to handle (as measured through the final clean and robust accuracy). This is true of BYORL as well as classical adversarial training. Exact analysis as to why this is the case and why that impacts BYORL differently from regular adversarial training remains to be done.
>
> > What is the effect of different transformations on robustness?
>
> Appendix B provides some insights. We evaluate the impact of color augmentation on BYORL and observe that the optimal color augmentation strength is slightly lower for BYORL than it is for standard BYOL. This can be explained by the fact that adversarial examples are a form of augmentation already and it is known that overly strong augmentations scheme hurt the performance of self-supervised techniques [3]. It is entirely possible that other transformations may impact BYORL differently.
>
> [1] Kim et al., "Adversarial Self-Supervised Contrastive Learning", 2020
> [2] Rice et al., "Overfitting in adversarially robust deep learning", 2020
> [3] Chen et al., "A simple framework for contrastive learning of visual representations", 2020

---

### Official Review · AnonReviewer1 · 2020-10-29
**Review report**

**Rating:** 6
**Confidence:** 4

**Review:**

In this paper, adversarial self-supervised learning is proposed to render robust data representations for down-stream fine-tuning tasks. The core idea is to integrate BYOL with adversarial training. The paper is well written in general. However, I do have several concerns about this submission.

1. Robust self-supervised pre-training + fine-tuning has been studied in two recent works at least.

[1] Hendrycks, Dan, et al. "Using self-supervised learning can improve model robustness and uncertainty." Advances in Neural Information Processing Systems. 2019.

[2] Chen, Tianlong, et al. "Adversarial Robustness: From Self-Supervised Pre-Training to Fine-Tuning." Proceedings of the IEEE/CVF Conference on Computer Vision and Pattern Recognition. 2020.

The comparison with [1-2] is recommended.

2. Unclear algorithm implementation.

a) In Figure 2, why is additional prediction head $q(\cdot, \theta)$ in BYOL and BYORL needed? Please clarify it.

b) It is not clear why a very large eps used in $\ell_2$ robust training, e.g., eps = 128/255.

c) In Table 2, it is not clear why $\ell_\infty$ robust pre-training results are missing?

3. Not convinced transfer results on unseen tasks.

In the paper, the authors claimed that "For both representations, we train a robust linear model using adversarial training (see subsection 3.1) with a different label availability on STL-10 and CIFAR-100 against `$\ell_\infty$ and `2 norm-bounded perturbations."

Thus, I supposed that during fine-tuning, the robust representation is frozen, and only the linear classifier is adversarially trained, correct?

If so, in Table 2, the results on Robust (BYORL) + Non-robust on CIFAR-10 seem very strong. It indicated that the standard partial fine-tuning is able to preserve robustness from self-supervised robust representation. However, [2] founds a different conclusion. Thus, it is important to provide additional explanations and comparisons for the achieved results.

I also suggest the authors check if the proposed defense yields obfuscated gradients, e.g., having a plot of robustness versus different attack strength eps during evaluation.

Lastly, if the pre-training task is conducted over CIFAR-10, can the results be further improved?

---

> ### Author Response · Authors · 2020-11-12
> **Transfer of robustness and clarifications**
>
> Thank you for the review. We address the possibility that our models are gradient obfuscated towards the end of this answer. Hopefully, this convinces the reviewer of the quality of our experimental results.
>
> > Robust self-supervised pre-training + fine-tuning has been studied in two recent works.
>
> Thank you for bringing [1] and [2] to our attention. We have added them to the related work section.
>
> [1] focuses on combining supervised adversarial training with a self-supervised task without building general representations (both tasks are trained in parallel). In our approach, we first build general representations that are robust and then fine-tune a linear model on top of these representations. We believe our results are complementary as they also highlight robustness transfer to multiple downstream tasks.
>
> [2] studies adversarial pre-training on self-supervised tasks (selfie, jigsaw, rotation). Models are for the most part trained on CIFAR-10 and evaluated on CIFAR-10 (or close variants). Fine-tuning is either partial or full. In the partial case, all ResNet blocks following the 3rd block are re-trained. While we also focus on the partial setting, we only train a linear model. To the contrary of [2], we also evaluate how our the resulting representations transfer to new tasks (STL-10 and CIFAR-100); and we also note that their approach does not preserve robustness with standard fine-tuning (as opposed to our method).
>
> > In Figure 2, why is additional prediction head $q(\cdot,\theta)$ in BYOL and BYORL needed?
>
> We would like to refer the reviewer to the original BYOL publication [3]. This head is needed by the online network to match the target network despite having to use different weights for the first two stage (as the target network is a slow moving averaging of the online network to avoid collapse).
>
>  > It is not clear why a very large eps used in l-2 robust training
>
> 128/255 and 0.5 are standard radii for l2 robustness [4, 5].
>
> > In Table 2, it is not clear why l-inf robust pre-training results are missing?
>
> We decided to split the different ablation experiments between l-2 and l-inf. There was no particular reason to focus l2 for this table (more experiments using l-inf are in the appendix).
>
> > I supposed that during fine-tuning, the robust representation is frozen, and only the linear classifier is adversarially trained, correct?
>
> Yes, this is correct.
>
> > the results on Robust (BYORL) + Non-robust on CIFAR-10 seem very strong.
>
> Thank you for highlighting one of the main results of this paper. We believe to be the first to demonstrate that, in practice, it is possible to transfer representational robustness to downstream tasks (without the need to perform robust fine-tuning). As indicated, this result is very strong and worth sharing.
>
> > However, [2] founds a different conclusion.
>
> As explained earlier in this answer, there is a rather large difference between our approach and the one in [2]. In [2], the authors fine-tune a deep non-linear model on top of their frozen representations, whereas we only fine-tune a linear model. We will make this distinction clear. In theory, we expect a drop of robust accuracy when using standard fine-tuning. In practice, that drop is relatively minor (at least on CIFAR-10 in the case of BYORL).
>
> > I also suggest the authors check if the proposed defense yields obfuscated gradients
>
> This was also our main worry. This is one of the reasons why all figures and tables show the robust accuracy resulting from one of the strongest combination of attacks (AutoAttack and MultiTargeted). We note that AutoAttack also uses a black-box attack (Square). We also ran the suggested evaluation (see below), which did not reveal further signs of gradient obfuscation. The loss landscapes do not exhibit any anomalies either (see Appendix).
>
> We evaluated the model corresponding to the 2nd row in Table 2 against l2 perturbations of size epsilon={1, 2, 3}. We also evaluated the same model against epsilon=128/255 with an increased number of PGD steps. For both evaluations, we combined AutoPGD with cross-entropy and difference of logits ratio losses [6].
> * eps = 1, number of steps = 100: Robust accuracy = 49.45%.
> * eps = 2, num steps = 100: Robust accuracy = 7.19%.
> * eps = 3, num steps = 100: Robust accuracy = 0%.
> * eps = 128/255, number of steps = 400: Robust accuracy = 77.27% (attack seem to have converged in 100 steps).
>
> > if the pre-training task is conducted over CIFAR-10, can the results be further improved?
>
> The pre-training task is already conducted over CIFAR-10 in Table 2. We do observe that robustness transfers well to unseen datasets (STL, CIFAR-100).
>
> [3] Grill et al., "Bootstrap your own latent: A new approach to self-supervised Learning", 2020
> [4] Rice et al., "Overfitting in adversarially robust deep learning", 2020
> [5] https://robustbench.github.io/
> [6] Croce and Hein. "Reliable evaluation of adversarial robustness with an ensemble of diverse parameter-free attacks", 2020

---

> > ### Comment · AnonReviewer1 · 2020-11-12
> > **Thanks for prompt response.**
> >
> > The authors have addressed some of my concerns except the following highlighted ones:
> >
> > (a) " We decided to split the different ablation experiments between l-2 and l-inf. There was no particular reason to focus l2 for this table (more experiments using l-inf are in the appendix). "
> >
> > I will really appreciate the authors to point out which table I should look into.
> >
> > I was expecting to see a similar supplementary table like Table 2 showing the 'robustness transferability' of \ell_\infty-adv-robust pre-trained BYORL.
> >
> > (b) Additional question based on Table 5 in Appendix, will ImageNet robust representation pretraining help? Compared to Table 2 using CIFAR-10 pre-trained network?
> >
> > (c) "As explained earlier in this answer, there is a rather large difference between our approach and the one in [2]. In [2], the authors fine-tune a deep non-linear model on top of their frozen representations, whereas we only fine-tune a linear model. We will make this distinction clear. In theory, we expect a drop of robust accuracy when using standard fine-tuning. In practice, that drop is relatively minor (at least on CIFAR-10 in the case of BYORL)."
> >
> > If the classification head is the root cause, then I would suggest having more results/empirical evidence to support it. Ablation studies on the choice of classification head or fine-tuning strategy (e.g., partial vs. full-network fine-tuning) would be nice.
> >
> > In summary, if authors can convince me that 1) transfer robustness is achieved by the proposal under different adversarial scenarios (\ell_infty, \ell_2, ImageNet-pre-training) and 2) Linear-fine-tuning is sufficient to preserve robustness， then it will enforce my review toward the positive side.

---

> > > ### Author Response · Authors · 2020-11-13
> > > **Additional results**
> > >
> > > Likewise, thank you for the quick response.
> > >
> > > > I will really appreciate the authors to point out which table I should look into. I was expecting to see a similar supplementary table like Table 2 showing the 'robustness transferability' of \ell_\infty-adv-robust pre-trained BYORL.
> > >
> > > We apologize for the misunderstanding: we meant that different experiments used different threat models (not necessarily the same experiment). In the meantime, we re-did Table 2 for l-inf perturbations of size 8/255.  The new l-inf table is in the revised manuscript (Table 6 in the appendix) and shows similar conclusions to Table 2. Note that a table dedicated to ImageNet is already there (Table 5). All results indicate that it is possible to transfer robustness across different adversarial scenarios.
> > >
> > > > Additional question based on Table 5 in Appendix, will ImageNet robust representation pretraining help? Compared to Table 2 using CIFAR-10 pre-trained network?
> > >
> > > This is an excellent question. In a bid to improve results against l-inf perturbations of size 8/255, we had tried the suggested experiment without much success on CIFAR-10. We were unable to obtain better representations when pre-training BYORL representations on ImageNet at epsilon = 8/255. There are two possible explanations:
> > > 1. It is possible that using a large epsilon (such as 8/255) is too destructive for ImageNet-sized images which contain a lot of details and textures. Prior work demonstrated that at 16/255, adversarial attacks are able to completely modify images (see Fig. 1 in [8]).
> > > 2. ImageNet might not be as useful for adversarial robustness on CIFAR-10 as additional images from the 80M Tiny Images dataset. Pre-training on ImageNet [7] is currently the lowest performing model using additional data on https://robustbench.github.io/.
> > >
> > > However, it is hard to draw any conclusions and we should re-evaluate this experiment with a smaller epsilon (4/255) or against l2 perturbations. It is likely that we will not have time to thoroughly perform this experiment before the end of the rebuttal period (as training robust ImageNet representations take close to 1 week), but we will strive to have it done as soon as possible.
> > >
> > > > If the classification head is the root cause, then I would suggest having more results/empirical evidence to support it. Ablation studies on the choice of classification head or fine-tuning strategy (e.g., partial vs. full-network fine-tuning) would be nice.
> > >
> > > The classification head is one issue (but not the only cause). That is, if we train a model $y = f \circ g (x)$ such that $y$ is robust to adversarial attacks on $x$, there is no guarantee that $z = g(x)$ will itself be robust. Robustness will depend on the loss/training used (e.g., in [2], representations stemming from the "rotation" pre-training are more robust under partial adversarial fine-tuning) as well as the architectures that parametrize $f$ and $g$. That being said, we performed the following experiment where instead of fine-tuning a linear head, we fine-tuned MLPs with 2 and 3 hidden layers (non-robustly and using the best pre-trained l-2 robust model):
> > > * Linear head: Robust accuracy of 78.22% (against l-inf perturbations and AutoPGD-100).
> > > * 2-hidden layers MLP (256 hidden units per layer): Robust accuracy of 71.8%
> > > * 3-hidden layers MLP (256 hidden units per layer): Robust accuracy of 70.9%
> > >
> > > These results demonstrate that using a deeper head reduces robustness when standard fine-tuning is used. The results, however, are quite encouraging, as non-trivial level of robustness are retained. This would indicate that BYORL itself has a greater ability to enforce that $z$ is robust and useful for downstream tasks than the approach in [2].
> > >
> > > Full-network finetuning reduces robust accuracy to 0% if trained for sufficiently long (as shown in [2]). This is in part explained by the use of standard cross-entropy which will continue to push correct logits as high as possible (at the expense of smoothness and ultimately robustness).
> > >
> > > [2] Chen et al. "Adversarial Robustness: From Self-Supervised Pre-Training to Fine-Tuning.", 2020.
> > > [7] Hendrycks et al., "Using Pre-Training Can Improve Model Robustness and Uncertainty", 2019
> > > [8] Zoran et al., "Towards Robust Image Classification Using Sequential Attention Models", 2019

---

> > > > ### Author Response · Authors · 2020-11-20
> > > > **Non-robust fine-tuning on ImageNet**
> > > >
> > > > The latest manuscript revision now contains updated results for ImageNet. Similarly to the previous results, we also observe that robustness transfers well when using non-robust fine-tuning.
> > > >
> > > > We sincerely appreciate your time and effort to review our paper. We believe we have addressed any remaining comment or question. If you have time, please indicate if there are any other concerns of yours which we have not addressed, we would be pleased to clarify those points.

---

> > > > > ### Comment · AnonReviewer1 · 2020-11-23
> > > > > **Thanks for the careful response.**
> > > > >
> > > > > After taking a close look at the authors' response, most of my concerns have well been addressed and I will increase my score to 6. Thanks for the great effort.
> > > > >
> > > > > My remaining concern is about ImageNet.
> > > > >
> > > > > "The latest manuscript revision now contains updated results for ImageNet."
> > > > >
> > > > > Which table/figure did you refer to?
> > > > >
> > > > > I believe that robust ImageNet pre-training for various downstream tasks is a more important and challenging question for adversarial defense. However, I understood "We were unable to obtain better representations when pre-training BYORL representations on ImageNet at epsilon = 8/255." Considering that SOTA was achieved under the CIFAR-10 pre-trained model, I believe that this submission has its own merits. This is my reason for increasing the score to 6.

---

### Official Review · AnonReviewer4 · 2020-10-29
**Application based approach, claims not very well supported.**

**Rating:** 4
**Confidence:** 4

**Review:**


This paper proposes some modifications to BYOL (Bootstrap Your Own Latents) in an attempt to address adversarial robustness in low-label high data regimes. The paper is well written and very easy to follow. Overall, the idea is clear and well presented.

Although the author's claim their approach to be novel, as is, the main contribution of the paper is adding adversarial training to BYOL, which already does not require labels.

The authors also claim that their approach is better than pseudo-labeling training for downstream tasks. Yet, this is not always the case. The authors also mention that their approach reaches optimal performance with 5% of labels (and seems to deteriorate with more labelled data). There is no analysis on why this is the case.

There is also a lack of comparison against contemporary SOTA self-supervised approaches.

---

> ### Author Response · Authors · 2020-11-12
> **Clarifications**
>
> Thank you for highlighting that the paper is easy to follow and clear.
>
> As correctly pointed out, we use adversarial training within BYOL. This requires making a few non-trivial choices (explained in Sec. 3.3). Additionally, we also perform a large number of experiments across different threat models and datasets and obtain conclusions that are worth sharing with the wider community. In particular:
> * We demonstrate that it is possible to obtain high robustness in the very low-label regime, see Fig. 3 and 4.
> * We also demonstrate that, contrary to popular belief [2], it is possible to preserve the robustness of self-supervised robust representations with standard partial fine-tuning (rather than robust fine-tuning), see Table 2.
>
> > The authors also claim that their approach is better than pseudo-labeling training for downstream tasks. Yet, this is not always the case.
>
> We are not sure what the reviewer is referring to when stating that BYORL is not always better than pseudo-labeling for downstream tasks. Table 1 shows that either BYORL matches pseudo-labeling (within less 0.5 percentage points except for a single case) or vastly outperforms it.
>
> > The authors also mention that their approach reaches optimal performance with 5% of labels (and seems to deteriorate with more labelled data).
>
> We meant that for all settings, BYORL does just as well with 5% of labels than with 100% of labels. The little fluctuations thereafter are due to using different initializations as models are re-trained from scratch for each setting (fluctuations are smaller than 0.5% which is not uncommon for adversarial training).
>
> > There is also a lack of comparison against contemporary SOTA self-supervised approaches.
>
> We compare BYORL to one of the best semi-supervised technique to train adversarially robust models (i.e., UAT) [1,4]. Except for [2], we are not aware of many self-supervised techniques that have been applied to adversarial training (at the time of submission). We did experiment separately with SimCLR but found that the resulting network were prone to gradient obfuscation. We would appreciate if the reviewer could provide some references.
>
> In [2], the authors do not focus on representation learning and, as such, they require deeper non-linear models when fine-tuning. To the contrary of our approach, their approach does not preserve robustness with standard fine-tuning. Finally, we also note that the authors of [2] used PGD-20 for their evaluation and, as such, results are not directly comparable. In fact, their best partially-finetuned model (which fine-tunes a much deeper model, but is the setting most similar to ours) reaches 45.10% robust accuracy against PGD-20 whereas we obtain 46.06% against AutoPGD-100 [3]. We have expanded our related work section in the new manuscript.
>
> [1] https://robustbench.github.io/
> [2] Chen, Tianlong, et al. "Adversarial Robustness: From Self-Supervised Pre-Training to Fine-Tuning." Proceedings of the IEEE/CVF Conference on Computer Vision and Pattern Recognition. 2020.
> [3] Croce and Hein. "Reliable evaluation of adversarial robustness with an ensemble of diverse parameter-free attacks", 2020
> [4] Uesato et al. "Are labels required for improving adversarial robustness?", 2019

---

### Author Response · Authors · 2020-11-20
**Thank you**

We would like to thank all reviewers for the time and effort spent reviewing this paper. We believe that we have addressed all concerns so far. However, if you have time, please indicate if there are any other concerns of yours which we have not addressed and we would be pleased to clarify those points.

Here is a summary of updates made to the manuscript:
* Improvements to the related work section which now includes more papers and highlights the differences with our approach more clearly (Sec 2).
* Clarifications on the algorithm and general improvements to the text (Sec 3.3)
* Clean accuracy is now reported in Fig. 7 and 8.
* More experiments on the the transfer of robustness without adversarial training (Table 6 and updated Table 5) .
* Updated ImageNet results which demonstrate improvements over classical adversarial training (Table 5).

Additionally:
* We addressed any concerns regarding gradient obfuscation (see answer https://openreview.net/forum?id=bgQek2O63w&noteId=4axSJPSrrqk).
* We performed an ablation study on the choice of classification head (see https://openreview.net/forum?id=bgQek2O63w&noteId=rZ18KBiedI6)

---

### Decision · Program_Chairs · 2021-01-07
**Final Decision**

**Decision:**

Accept (Poster)

**Comment:**

The paper considers the use of adversarial self-supervised learning to render robust data representations for various tasks, in particular to integrate the Bootstrap Your Own Robust Latents (BYOL) with adversarial training, where a small amount of labeled data is available together with a sizable unlabeled dataset.  Especially the low-data regime is of interest.  It extends a previous method with a new adversarial augmentation technique, it is compared against several methods, and the robust representations are shown to be useful more generally.  There were some confusing presentations and questions that were resolved in a detailed discussion with the reviewers.